# A Novel Mechanism for Transcription Termination in the *mod(mdg4)* Locus of *Drosophila melanogaster*

**DOI:** 10.3390/biology13120994

**Published:** 2024-11-29

**Authors:** Iuliia V. Soldatova, Mikhail V. Shepelev, Pavel Georgiev, Maxim Tikhonov

**Affiliations:** 1Department of Regulation of Genetic Processes, Institute of Gene Biology Russian Academy of Sciences, Moscow 119334, Russia; nao.jem@gmail.com; 2Center for Precision Genome Editing and Genetic Technologies for Biomedicine, Institute of Gene Biology Russian Academy of Sciences, Moscow 119334, Russia; mshepelev@mail.ru

**Keywords:** transcription termination, RNA processing, RNA secondary structure, trans-splicing, *mod(mdg4)*, *Drosophila melanogaster*

## Abstract

The expression of protein-coding genes in eukaryotes ends with cleavage and polyadenylation, followed by the termination of transcription. Previously, we identified a non-canonical transcription terminator (NTT) in the *mod(mdg4)* gene of *Drosophila melanogaster* that causes transcription to stop, producing an mRNA that lacks a poly(A) tail. Here, we identified a minimal functional unit of the NTT consisting of 79 nucleotides that form a specific secondary RNA structure. Transcripts generated from the NTT demonstrated reduced stability and were impeded in their export to the cytoplasm. Notably, the NTT did not exhibit functionality in human cells, indicating that this termination mechanism is not universal and necessitates interaction with specific proteins. Interestingly, an NTT from the distantly related species *D. willistoni* could effectively terminate transcription in *D. melanogaster* cells, highlighting the importance of conserved motifs for NTT functionality. Overall, these findings provide insights into alternative mechanisms of transcription termination and underscore the significance of RNA primary sequences and secondary structures in potential interactions with specific proteins.

## 1. Introduction

Transcription can be divided into three phases: initiation, elongation, and termination. Transcription terminates after RNA polymerase II synthesizes the coding region of a gene and reaches the polyadenylation signal (PAS). This process involves the cleavage of a pre-mRNA and its conversion into a mature transcript through polyadenylation at the 3′ end (the poly(A)-dependent mechanism). Splicing and transcription termination are mutually regulated and coupled with the transcription of pre-mRNA [1,2,3].

In addition to the classical termination mechanism, several alternative mechanisms have recently been discovered in animals, and these are typically associated with the formation of non-coding RNAs and involve the Integrator complex [4], the ZC3H4-WDR82 restriction complex [5], and the microprocessor [6]. In classical transcription termination, the cleavage and polyadenylation of the pre-mRNA are carried out by a large multi-protein complex that is associated with RNA polymerase II [7]. During transcription elongation, splicing factors inhibit the activity of the PASs located within introns [8,9,10].

We previously described a conserved sequence in the intron of the *mod(mdg4)* locus that blocks transcription [11]. The *mod(mdg4)* locus encodes more than 30 isoforms that differ in their C-terminal domains. The mRNAs encoding *mod(mdg4)* isoforms are generated only through trans-splicing [11,12,13]. The identified motifs necessary for trans-splicing are located within the fourth intron, which separates the constitutive exons of the gene from the alternative exons. A conserved sequence forming a stable secondary RNA structure was identified in the center of the fourth intron [13] (Figure 1A).

In this central region of the intron, there is a pause in the action of RNA polymerase II, leading to a complete interruption of transcription. The resulting transcripts lack a poly(A) tail, and the transcription of individual pre-mRNAs may terminate at extended regions without a precise boundary (Figure 1A) [11]. This sequence has been termed the “non-canonical transcription terminator” (NTT). Termination within the last constitutive intron prevents transcription readthrough into the downstream region where alternative exons are located. This intronic transcriptional arrest prevents cis-splicing, resulting in the transcription of all alternative exons through separate promoters. This mechanism may be crucial for the balanced production of different isoforms solely through trans-splicing.

A fragment of 763 nucleotides, including the NTT, effectively terminates transcription from the actin promoter when positioned within the intron of the yellow gene in a transgenic construct. This 763-nucleotide fragment induced termination in a model system during transient transfection of S2 cell cultures. The RNA terminated at the NTT could not be exported from the nucleus to the cytoplasm [11].

The present study aimed to determine the minimal functional unit of the NTT, investigate the conservation of its function within the *Drosophilidae* family, and analyze the ability of the NTT to induce the termination of transcription in human cells.

## 2. Materials and Methods

### 2.1. Genetic Constructs

Vectors were generated using standard molecular biology techniques using enzymes purchased from Thermo Fisher Scientific (Waltham, MA, USA). Plasmids were produced in the *Escherichia coli DH5α* strain. The constructs were based on the pAc5.1/V5-His vector (Thermo Fisher Scientific, Waltham, MA, USA). The *Fluc* and *Rluc* genes were taken from the pGL3-Basic and pRL vectors (Promega Corporation, Madison, WI, USA), respectively. The NTT sequences were amplified from the genomic DNA of *Drosophila melanogaster*. A synthetic element and a fragment from *Drosophila willistoni* were obtained using a set of overlapping oligonucleotides. A detailed scheme of the cloning procedures, the vectors, and the oligonucleotides used is given in Appendix A.

### 2.2. Culturing and Transfection of S2 Cells

S2 cells were cultured in SFX medium at 25 °C. In brief, 8 × 10^5^ cells were seeded into six-well plates and co-transfected with 200 ng of the pAc-Fluc plasmid and 800 ng of the experimental plasmid using the Cellfectin II reagent (Thermo Fisher Scientific, Waltham, MA, USA) following the manufacturer’s instructions. After transfection, the cells were grown for 48 h and then were washed with 2 mL of PBS and divided into two samples, comprising 800 µL for a luciferase analysis and 1200 µL for RNA extraction.

### 2.3. Culturing and Transfection of HEK293 Cells

HEK293 cells (European Collection of Cell Cultures, 85120602) were cultured in DMEM (SH30285.01, Cytiva, Wilmington, DE, USA) supplemented with 10% fetal bovine serum (SV30160.03, Cytiva, Wilmington, DE, USA), 2 mM L-glutamine, 100 U/mL penicillin, and 100 µg/mL streptomycin. The cells were maintained at 37 °C in a humidified atmosphere containing 5% CO_2_. For RNA extraction, 500,000 HEK293 cells were seeded into a six-well plate with 2 mL of complete medium. The next day, the cells were transfected with 2 µg of plasmid DNA per well using the DreamFect transfection reagent (OZ Bioscience, France). After 48 h, the cells were washed with PBS, and 1 mL of TRI reagent (MRC, Cincinnati, OH, USA) was added for RNA extraction.

### 2.4. Drosophila Stocks and Husbandry

*Drosophila melanogaster* lines were maintained at 25 °C under standard cultivation conditions. Transgenic constructs were integrated into the 22A2 chromosomal region using φC31 integrase. All of the constructs were based on a vector containing an attB site and the *white* gene. The vector DNA, at a concentration of 200 ng/μL, was microinjected into the posterior end of fly embryos [14] of a line with an attP site and the φC31 integrase gene, controlled by the vasa gene promoter (line #24481 from the Bloomington Drosophila Stock Center, Genotype: *y^1^ M{vas-int.Dm}ZH-2A w^1118^; M{3xP3-RFP.attP’}ZH-22A*) [15]. The resulting flies from the microinjection were crossed with the white-eyed laboratory line *y*^1^*w*^1118^, and the transgenic offspring were identified through phenotypic expression of the *white* gene, which determines eye pigmentation. Homozygous lines were bred using a second chromosome balancer line (*y*^1^*w*^1118^; *Kr*^If−1^/Sm5). The insertions of the constructs were verified through PCR using the following primer pair that amplified the junction between the construct and the genome:attB_d GTCGACGATGTAGGTCACGGand loc22A_r ACCTCGGAATGAAGGAGTGAAG.

### 2.5. The Luciferase Assay

Following transfection, the cells were collected and pelleted through centrifugation (400 g, 2 min). The resulting pellet was lysed, and the luciferase activity in the lysate was measured using a Firefly and Renilla Luciferase Assay Kit (Biotium, Inc., Fremont, CA, USA), according to the manufacturer’s instructions. Each transfection and the measurements for each construct were replicated at least three times.

### 2.6. Quantification of Gene Expression by qRT-PCR

The transfected S2-cells were pelleted through centrifugation (400 g, 2 min), and 1 mL of TRI Reagent^®^ (MRC, Cincinnati, OH, USA) was subsequently added to the pellet. The HEK293 cells were washed with PBS directly in the well, and 1 mL of TRI reagent^®^ (MRC, Cincinnati, OH, USA) was added. For each biological replicate, 15–30 adult males aged 2 to 3 days old were frozen in liquid nitrogen, and 200 µL of TRI Reagent^®^ (MRC, Cincinnati, OH, USA) was added to each sample. The samples were then homogenized with a pestle, and the final volume was adjusted to 1 mL. RNA extraction was performed according to the manufacturer’s protocol, and the RNA samples were treated with DNase (Thermo Fisher Scientific, Waltham, MA, USA) to eliminate any residual genomic DNA. cDNA synthesis was performed using RevertAid (Thermo Fisher Scientific, Waltham, MA, USA) reverse transcriptase in a reaction mixture containing 5 μg of RNA and 25 μM random hexamers. The resulting cDNA samples were analyzed via quantitative PCR using the polymerase and buffers from a Diamant TaqA kit (BelBioLab, Moscow, Russia). EvaGreen (Biotium, Inc., Fremont, CA, USA) was used as the intercalating dye, while Rox (Biotium, Inc., Fremont, CA, USA) served as the reference dye. Amplification was performed using the QuantStudio™ 6 Flex Real-Time PCR System (Thermo Fisher Scientific, Waltham, MA, USA). The primers used for the analysis were as follows: Rluc_RT_d CAGTGGTGGGCCAGATGTAAACAA, Rluc_RT_r TAATACACCGCGCTACTGGCTCAA, eGFP_RT_d AGCAGAAGAACGGCATCAAG, eGFP_RT_r GGTGCTCAGGTAGTGGTTGTC, Fluc_RT_d TTGCTCCAACACCCCAACAT, and Fluc_RT_r TTCCGTGCTCCAAAACAACA. The relative levels of mRNA expression were calculated within the linear amplification range by calibrating them to a standard curve. Serial dilutions of a plasmid containing three genes, *Rluc*, *Fluc*, and *eGFP*, were used as the standards. The significance of changes in the expression levels was assessed using Student’s *t*-test.

## 3. Results

### 3.1. Model System for Testing DNA Sequences for Their Ability to Initiate Transcription Termination

To assess the ability of the tested DNA fragments to initiate transcription termination, we used a vector containing two reporter genes: *Rluc* (luciferase from *Renilla reniformis*) and *eGFP* (an enhanced green fluorescent protein). Both reporter genes were located within a single transcription unit under the control of the Act5C promoter (Figure 2A). This vector (Act-Rluc-eGFP) with a strong promoter is optimal for inducing expression in *Drosophila* S2 cell cultures. The tested DNA fragments were inserted between *Rluc* and *eGFP*. If a DNA fragment terminated transcription, there was a decrease in the *eGFP* levels relative to those of *Rluc* that was determined using quantitative PCR. The effectiveness of the elements was compared to the strong polyadenylation signal from the SV40 virus. As a control for transfection efficiency, a plasmid containing the *Fluc* gene (firefly luciferase) under the control of the Act5C promoter was co-transfected with the target plasmid.

### 3.2. Identification of the Minimal DNA Fragment Capable of Inducing Transcription Termination

Deletion derivatives of the original 763 bp DNA fragment were obtained to identify functionally important elements (Figure 2B). The conserved region (the core DNA fragment) that formed a secondary structure in the RNA was divided into smaller fragments (603–723 and 723–775) as the most likely candidates that terminate transcription. The least conserved part of the fourth intron, a sequence that is rich in simple repeats, was located downstream from the core region. This was designated as a 412 bp fragment (786–1198). At the 3′ end, there was a 168 bp fragment with several conserved sequences among the *Drosophilidae* family. Each modified DNA fragment was integrated into the Act-Rluc-eGFP vector to quantify the efficiency of transcription termination. In the control construct without the DNA fragments, the *eGFP*/*Rluc* ratio was close to 1. This was consistent with the absence of termination between the reporter genes (Figure 2B). In contrast, there was a significant decrease in the *eGFP*/*Rluc* ratio, to 0.07, for the control construct with the SV40 polyadenylation signal, indicating effective termination of transcription between the reporter genes. When the S2 cells were transfected with vectors carrying sequences containing the secondary structure, there was a significant decrease in the *eGFP*/*Rluc* ratio, ranging from 0.35 to 0.24, suggesting termination of transcription (Figure 2B). Notably, the observed transcription termination was not as efficient as in the presence of the SV40 PAS. Moreover, deletion of the DNA sequences located upstream and downstream of the secondary structure did not affect the efficiency of transcription termination. Deletion of a small part of the secondary structure (Δ723–786) resulted in an *eGFP*/*Rluc* ratio corresponding to the level of the control construct without the elements, indicating the absence of transcription termination.

Our data demonstrate that the core sequence that forms the secondary structure of the RNA is solely responsible for the termination of transcription, as deletion of the flanking regions had no effect. To define the minimal functional element, two truncated variants of the core DNA fragment were created and tested: a 115 bp fragment encompassing the secondary structure and adjacent regions and a 79 bp fragment containing only the secondary structure. Both variants successfully terminated transcription, demonstrating that the 79 bp element comprising the RNA secondary structure was sufficient for this process. The minimal DNA sequence was designated as the NTT.

### 3.3. Transcripts Resulting from Non-Canonical Transcription Termination Exhibit Reduced Stability and Translation Efficiency

To assess the impact of non-canonical transcription termination on transcript steady-state levels, we measured the *Rluc*/*Fluc* mRNA ratio using *Fluc* from a co-transfected independent vector as an internal control (Figure 2D). We also quantified the luciferase activities of both genes (Figure 2E). The ratio of luciferase activity reflects the changes in the translation efficiency in the presence of the NTT.

The *Rluc* levels in the constructs containing a complete NTT were decreased compared to those in the control Act-Rluc-eGFP construct, with the reduction ranging from 1.5- to 3-fold (Figure 2D). In contrast, the *Rluc*/*Fluc* ratio was increased 2.8-fold in the construct with the SV40 PAS, while the construct with a deleted secondary structure portion showed a similar ratio to that of the control. This indicated that the NTT-derived RNA lacking a poly(A) tail was less stable, leading to reduced levels. Conversely, the short transcript processed by the SV40 PAS was more stable than that from the control construct containing both the *Rluc*- and *eGFP*-coding regions.

A comparison of the luciferase activity ratios (Figure 2E) showed an even more striking effect, with a decrease ranging from 2.5- to 10-fold relative to the control construct. This suggests that transcripts lacking a poly(A) tail may be hindered or be unable to be exported to the cytoplasm [11]. The observed Rluc luciferase activity likely resulted from the translation of mRNA that escaped termination via the NTT and was processed at the downstream polyadenylation site. Unexpectedly, a construct with a deleted NTT secondary structure fragment for which non-canonical transcription termination was not observed also exhibited reduced Rluc luciferase activity. This may have been due to the influence of other elements within the tested fragment on the translation efficiency.

### 3.4. The Secondary Structure Causes Transcription Termination in a Transgenic Drosophila Model System

Transient expression in S2 cells using reporter constructs did not ensure the effectiveness of the NTT in the genome. Therefore, the next objective was to investigate the ability of the NTT to mediate termination upon genomic integration. For this purpose, a transgenic Drosophila line with an attP site recognized by φC31-integrase in the 22A cytogenetic locus was used [15]. Control (Act-Rluc-eGFP) and derivative constructs containing two NTT variants (137 and 137 + 168) and the SV40 PAS were transferred into a vector with an attB site and the mini-white reporter gene for transgene identification. Construct 137 + 168 was created because the 168 region was previously shown to be necessary for termination in a model system that included the complete intron 4 of the *mod(mdg4)* gene [11]. Homozygous lines were established, and the *eGFP*/*Rluc* ratio was measured using qPCR in two- to three-day-old males (Figure 3A). The NTT lines exhibited a decreased *eGFP*/*Rluc* ratio compared to that of the control, indicating relatively efficient termination of the transcription between the *eGFP* and *Rluc* genes, although this was less effective than that of the SV40 PAS. Notably, both the NTT and the SV40 PAS showed enhanced efficiency when integrated into the genome compared to when transiently expressed in S2 cells.

In conclusion, the NTT functions more effectively when integrated into the genome than when it is expressed from plasmids in S2 cells.

### 3.5. The Drosophila NTT Does Not Induce the Termination of Transcription in Human Cells

Our findings suggested that the NTT’s secondary structure may have a conserved evolutionary role in non-canonical transcription termination. To test this hypothesis, we investigated whether the *Drosophila* NTT could terminate transcription in human HEK293 cells. We generated experimental constructs containing the NTT and two control constructs (Rluc-eGFP and SV40 PAS) under the control of the CMV promoter that is active in human cells. The HEK293 cells were transfected with these plasmids using standard techniques. Three independent experiments revealed that the *eGFP*/*Rluc* ratio for the NTT constructs was similar to that of the control pCMV-Rluc-eGFP, while the construct containing the SV40 PAS showed a decreased ratio, indicating transcription termination (Figure 3A). These results demonstrate that the NTT does not possess universal activity and that non-canonical transcription termination requires specific protein factors absent in human cells.

### 3.6. The Primary Sequence of the NTT Is Essential for Transcription Termination in D. melanogaster

NTT RNA forms a secondary structure that likely plays a crucial role in the termination of transcription. Moreover, the primary sequence of the NTT is highly conserved among *Drosophilidae* and may specifically interact with protein factors involved in termination. To evaluate the role of the primary sequence, a synthetic 79 bp sequence was designed to closely mimic the secondary structure of the NTT RNA (Figure 4A), maintaining the positions and lengths of the internal loops, hairpins, and stems. This synthetic sequence was integrated into the Act-Rluc-eGFP vector between the two marker genes and used to transfect *Drosophila* S2 cells.

As demonstrated by the *eGFP*/*Rluc* ratio measurements (Figure 4B), the synthetic element failed to terminate transcription, underscoring the importance of the primary nucleotide sequence of the NTT element.

### 3.7. The NTT Element of D. willistoni Functions as a Transcription Terminator in D. melanogaster

The NTT exhibited high primary sequence conservation across *Drosophila* species, with *D. willistoni* showing the most significant differences. Both species share three conserved regions among *Drosophilidae*. Two of these regions are located at the 5′ and 3′ ends, forming the hairpin base (Figure 4A), while the third region from the center of the NTT forms a loop and its supporting stem. The divergent sequences between these distant species form the main stems of the secondary structures. Notably, the nucleotide sequence of the *D. willistoni* NTT (dwNTT) was the longest among the analyzed species. Additionally, the non-conserved sequences unique to this species (Figure 4A) are mutually complementary, further elongating the overall length of the stem. As in previous experiments, the *D. willistoni* DNA fragment was inserted between *Rluc* and *eGFP*. The resulting expression vector was transfected into S2 cells along with a control construct lacking any elements. The decreased *eGFP*/*Rluc* ratio indicated that the dwNTT could support non-canonical transcription termination in the *D. melanogaster* cells (Figure 4B). These findings demonstrate that the dwNTT induced transcription termination in the model system. A comparison of the *Drosophila melanogaster* and *Drosophila willistoni* sequences revealed only three conserved regions. The sequences connecting these regions show significant divergence. However, the termination efficiency of the dwNTT is comparable to that of the NTT. We therefore propose that the small conserved regions that form the hairpin and its base likely serve as key motifs recognized by the proteins involved in non-canonical transcription termination.

## 4. Discussion

In this study, we demonstrated that a 79 bp NTT sequence is sufficient to induce the termination of transcription. The NTT RNA forms a secondary structure that likely plays a functional role in this process. Notably, NTT-mediated termination appears to be independent of canonical polyadenylation and occurs without poly(A) tail formation, resulting in reduced transcript stability and impaired cytoplasmic transport. Unlike classical termination, which is repressed in introns, the NTT can induce transcription termination within gene introns.

In the context of the *mod(mdg4)* gene, the function of the NTT may be to promote intragenic termination, thereby preventing cis-splicing and preserving the donor splice site for trans-splicing with acceptor transcripts. This mechanism ensures that all *mod(mdg4)* isoforms are generated exclusively through trans-splicing [11]. In a similar way, alternative splicing is regulated by canonical polyadenylation signals [16,17].

Two potential mechanisms for non-canonical transcription termination have been proposed. The first mechanism posits that the RNA structure acts as an aptamer for components of the elongation complex, with the RNA secondary structure alone being sufficient for termination. The second mechanism involves protein partners that bind to the RNA and inhibit transcription by RNA polymerase II. Our results indicate that a synthetic structure with a similar form does not induce termination and that the NTT does not function in human cells. In contrast, the homologous structure from the distantly related species *D. willistoni* supported termination in *D. melanogaster*, suggesting the involvement of proteins specific to *Drosophilidae* that recognize conserved motifs in the corresponding NTT RNA.

Thus, our findings support a model in which the functions of the NTT are determined by both the RNA’s secondary structure and specific motifs recognized by particular proteins. The slowing and pausing of RNA polymerase II followed by the termination of transcription may result from the combined action of the RNA structure, as this could reduce the speed of RNA polymerase II’s movement and enhance the binding efficiency of termination-associated proteins.

Earlier studies on trans-splicing attempted to identify the proteins that bind to intron 4 [13]. For the in vitro-transcribed RNA of the region where we identified the NTT, a pool of proteins has been characterized, which includes components of the classical polyadenylation machinery (Symplekin, CPSF 160, and PABP), ATP-dependent RNA helicases (MLE and me31b), and ubiquitous proteins (Pep, histone H4, and H2A/e), among other things. Further study is needed to identify and validate the RNA-binding proteins responsible for NTT-mediated termination of transcription.

In situations where transcription must be terminated without the production of mRNA, non-canonical transcription termination exhibits characteristics that confer advantages over polyadenylation signals. In contrast to the NTT, the activity of the PAS is inhibited by the spliceosome, which is responsible for the splicing of gene introns. Transcripts that lack poly(A) tails exhibit a reduced export capacity and stability, thereby preventing the formation of aberrant proteins. It is plausible to speculate that NTT-like elements may be involved in the generation of RNAs that function within the nucleus. Furthermore, it is conceivable that transcription termination signals analogous to those of the NTT may be widespread in the genomes of various animal species, including mammals.

## 5. Conclusions

Our study elucidates a novel mechanism of transcription termination through the identification and characterization of a non-canonical transcription terminator in the *mod(mdg4)* gene of *Drosophila melanogaster*. We demonstrated that a minimal 79 bp sequence, capable of forming a specific secondary RNA structure, is sufficient to mediate transcription termination independently of polyadenylation. The primary sequence of the NTT is essential for transcription termination in *D. melanogaster*. Furthermore, the ability of the NTT from the distantly related *Drosophila willistoni* to induce transcription termination in *D. melanogaster* highlights the importance of conserved motifs.

## Figures and Tables

**Figure 1 biology-13-00994-f001:**
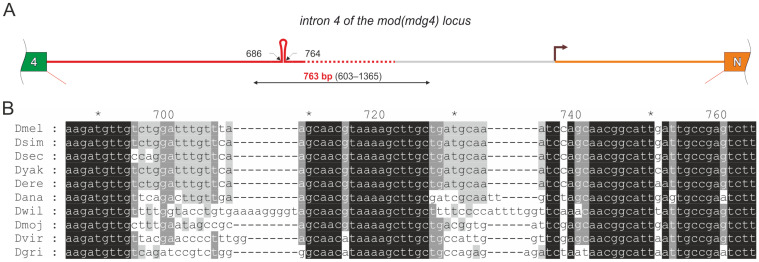
(**A**) The structure of the fourth intron of the *mod(mdg4)* gene in *Drosophila melanogaster*. Non-canonical termination of transcription occurs within this intron, producing transcripts that lack a poly(A) tail. The resulting pool of transcripts does not exhibit a defined boundary at the 3′ end (indicated by the red dashed line). The gray line denotes the region where transcription is absent. The 763 bp fragment (indicated by the black double arrow) is responsible for inducing non-canonical termination of transcription [11]. A conserved sequence that forms a secondary structure occurs within the region 686–764. The fourth constitutive exon is marked in green. The annotated promoter of the alternative acceptor exon N (shown in orange) is indicated by the arrow. (**B**) Alignment of the core sequence from 686 to 764 bp with homologous fragments of introns from various species within the *Drosophilidae* family. The top row provides coordinates relative to the start of intron 4 in *D. melanogaster*, with every 10th position marked by a number or an asterisk (*).

**Figure 2 biology-13-00994-f002:**
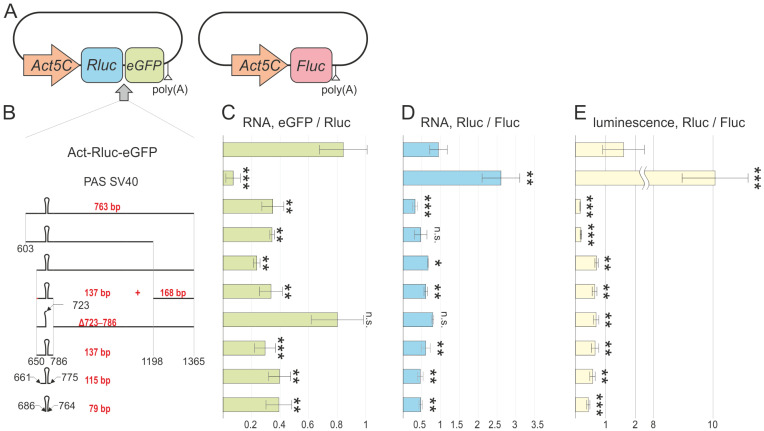
(**A**) A model system for evaluating the ability of DNA sequences to initiate the termination of transcription. The *Rluc* and *eGFP* genes are located within the same transcription unit under the control of an actin promoter. DNA fragments were inserted between the *Rluc* and *eGFP* genes for assessment. A plasmid containing the *Fluc* gene under the control of the Act5C promoter was co-transfected with the experimental construct for normalization. (**B**) Schematic representations of deletion derivatives of the original fragment assessed for their ability to induce transcription termination. A control vector without the elements (Act-Rluc-eGFP) and with the SV40 polyadenylation signal was used. The ratios of the transcript levels of eGFP to *Rluc* (**C**) and *Rluc* to *Fluc* (**D**) and the luminescence ratios of Rluc to Fluc (**E**) were measured. Each measurement was conducted at least three times, and the error bars represent standard deviations. The asterisks indicate the levels of statistical significance, ** *p* < 0.01 and *** *p* < 0.001, non-significant (n.s.) *p* > 0.05, with comparisons made against the control construct Act-Rluc-eGFP.

**Figure 3 biology-13-00994-f003:**
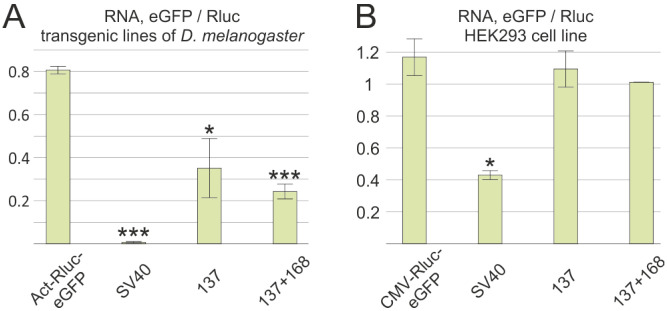
Evaluation of two fragments containing an NTT for their ability to induce the termination of transcription (**A**) within the *D. melanogaster* genome and (**B**) in human HEK293 cells. The ratios of *eGFP* to *Rluc* transcript levels are presented. Each measurement was conducted at least three times, and the error bars represent standard deviations. The asterisks denote levels of statistical significance, * *p* < 0.05, *** *p* < 0.001, with comparisons made against the control construct Rluc-eGFP.

**Figure 4 biology-13-00994-f004:**
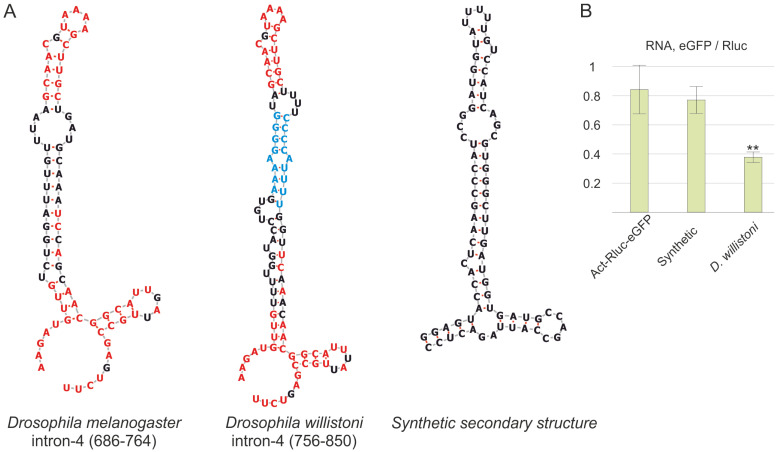
(**A**) RNA secondary structures of the NTT from *D. melanogaster* and *D. willistoni* and an artificial synthetic structure. Conserved regions among *Drosophilidae* (highlighted in red) form the hairpin base, loop, and supporting stem. Non-conserved regions of the RNA structure in *D. willistoni* are shown in blue. (**B**) Evaluation of the artificial synthetic structure and the NTT from *D. willistoni* for their ability to induce the termination of transcription in a model system. The ratio of *eGFP* to *Rluc* transcript levels is shown. Each measurement was conducted at least three times, and the error bars represent standard deviations. The asterisks indicate levels of statistical significance, ** *p* < 0.01, with comparisons made against the control construct Act-Rluc-eGFP.

## Data Availability

Inquiries can be directed to the corresponding author.

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
