# Peer review of "A Novel Mechanism for Transcription Termination in the mod(mdg4) Locus of Drosophila melanogaster"

_biology, 2024, doi:10.3390/biology13120994_

Round 1

Reviewer 1 Report

Comments and Suggestions for Authors

Soldatova et al. has previously reported that a fragment of 763 bp serves as a non-canonical transcription termination (NTT) site in the mod locus of Drosophila and aims to identify the minimal region that functions as the transcription terminator. To this extent, the authors first establish two plasmid constructs, one as a fusion construct (RLuc-eGFP) and the other being FLuc, both of which, are expressed under the control of a constitutive Act5C promoter. Then the authors use a series of deletion constructs, along with SV40 PAS as the positive control, and check the ability of the constructs to trigger transcription termination in the fusion construct. Interestingly, a minimal stem-loop structure of 79 bp is sufficient to facilitate transcription termination. However, the resulting transcripts are relatively less stable and inefficiently translated compared to the polyadenylated products generated from the SV40 construct. The authors then establish transgenic fly lines to test the ability of two of these constructs, namely 137 and 137 + 168, to induce transcription termination and they report that both constructs are functional in vivo in Drosophila but not in HEK293 cells, suggesting species specificity. Lastly, the authors test the ability of a random synthetic RNA with a similar secondary structure to function as the terminator, which it does not.

NTT is an interesting transcription termination mechanism, which has not been fully illuminated. I believe that this study contains well-constructed approaches to identify the minimal functional region sufficient to induce transcription termination. The deletion constructs are highly convincing and supported by in vivo data. I suggest the following suggestions to improve the manuscript:

1. Methods should be detailed such that they can be easily reproduced by others. For example, lines 88-95: What is being cloned in to which restriction sites.. etc. Line 104: How many cells? Line 139: Centrifugation details and many others.

2. Figure 2B: What is meant by dots? Please state in the figure caption.

3. Line 254-255: Incomplete sentence.

4. Figure 3: Although the authors identified a minimal sequence of 79 bp, they used longer constructs in Drosophila. Is there a specific reason for this? Why these two constructs?

5. Lines 356-362: I wonder if there is any PAR-CLIP-seq data in Drosphila that might provide insight into potential protein partners of this sequence.

Author Response

We thank the reviewer for the positive assessment of our work.

Comments 1: “Methods should be detailed such that they can be easily reproduced by others. For example, lines 88-95: What is being cloned in to which restriction sites.. etc. Line 104: How many cells? Line 139: Centrifugation details and many others.”

Response 1: We have tried to improve the Methods section. The detailed cloning sequence is described in Supplementary table 1. We have also made additions and clarifications in the Methods section to simplify reproducibility.

Comments 2: “Figure 2B: What is meant by dots? Please state in the figure caption.”

Response 2: We apologize, we missed this label on the Figure 2B. The dots indicate the same as in Figure 1. In the described non-canonical termination of transcription, there is no clear boundary where transcription ends. Previously, we examined transcription levels in various ways and observed that in the area marked by dots, the RNA level gradually decreases to zero. This means that the formation of the 3' end occurs in a fairly large region. However, we decided to remove the dot designation from the figure, since in this case it does not carry important information, but only confuses readers.

Comments 3: “Line 254-255: Incomplete sentence.”

Response 3: This sentence has been corrected.

Comments 4: “Figure 3: Although the authors identified a minimal sequence of 79 bp, they used longer constructs in Drosophila. Is there a specific reason for this? Why these two constructs?”

Response 4: We obtained  transgenic flies before it became clear which element is minimal. That is why construct 137 was created. Additionally, construct 137 + 168 was made because the 168 region was previously necessary for termination in a model system that includes the full intron 4 of the mod(mdg4) gene. At that time, we did not know whether the 168 fragment was needed for termination. Therefore, such constructs were made. Nevertheless, these constructs address the question of whether termination occurs in a heterologous construct within a genomic context.

Comments 5: “Lines 356-362: I wonder if there is any PAR-CLIP-seq data in Drosphila that might provide insight into potential protein partners of this sequence.”

Response 5: We thank the reviewer for this important question. It directly pertains to the issue we are currently interested in – partner proteins. Previously, in the work of Gao et al. 2015, proteins that bind to the fragment of intron 4, including NTT, were identified. We are now investigating which of these proteins may be involved in transcription termination. Appropriate additions have been made to the text

Reviewer 2 Report

Comments and Suggestions for Authors

This manuscript examined a non-canonical transcription terminator (NTT) in Drosophila melanogaster, which functions through a 79-nucleotide RNA structure. The NTT causes reduced transcript stability and export to the cytoplasm, and does not work in human cells. It is conserved across Drosophilidae, as shown by its functionality in D. willistoni cells. Here are some suggestions below.

1. In introduction, the background information about the mod(mdg4) locus and its role in generating multiple isoforms via trans-splicing is interesting but could be presented more concisely. Since the study focuses on identifying the minimal functional unit of the NTT, it may be more useful to provide a brief sentence early on explaining the role of NTT in mod(mdg4) gene regulation, especially in terms of how it prevents cis-splicing and promotes trans-splicing. Adding the importance of non-noncanonical termination in the term need might enhance the readers interests.

2. Although the author has included a detailed methodology, if the methods were not developed by the author, adding references could strengthen the presentation of the methodology section.

3. The model of NTT-mediated transcription termination involving RNA structure and protein binding is proposed in the discussion, but the Results section should provide more details about whether efforts have been made to identify the RNA-binding proteins involved. Including information about any planned or preliminary experiments would make the proposed model stronger and more grounded in experimental data.

4.   In discussion, author suggested that NTT plays a role in regulating splicing in the mod(mdg4) gene. It would be more valuable to add experimental evidence that links NTT to splicing or to discuss relevant literature that supports this function.

5. The comparative analysis with D. willistoni offers an interesting evolutionary perspective, but the discussion could further explore how sequence variations between species influence the functional outcome of NTT activity. Are the conserved regions alone sufficient for termination, or do the non-conserved variations play a role in termination efficiency? A more detailed discussion of these sequence differences would add depth to the evolutionary context of the study.

6.  While the proposed mechanism of non-canonical transcription termination is convincing, the Discussion should broaden its scope by connecting these findings to larger themes in transcription regulation. Adding the experimental evidences in the context of known non-canonical termination mechanisms in other species would underscore the importance of the findings in the field of transcription regulation.

7. Overall, the manuscript presents an interesting result. Including suggestions for future directions would further enhance its impact and provide valuable insights for similar future research.

Author Response

Comments 1: “In introduction, the background information about the mod(mdg4) locus and its role in generating multiple isoforms via trans-splicing is interesting but could be presented more concisely. Since the study focuses on identifying the minimal functional unit of the NTT, it may be more useful to provide a brief sentence early on explaining the role of NTT in mod(mdg4) gene regulation, especially in terms of how it prevents cis-splicing and promotes trans-splicing. Adding the importance of non-noncanonical termination in the term need might enhance the readers interests.”

Response 1: We thank reviewer for the suggestion. We added our explanation for potential role of noncanonical termination in the fourth intron.

Comments 2: “Although the author has included a detailed methodology, if the methods were not developed by the author, adding references could strengthen the presentation of the methodology section.”

Response 2: At the request of the reviewer, we have indicated the manufacturers of the enzymes and kits used. The reference to the microinjection method has also been added.

Comments 3: “The model of NTT-mediated transcription termination involving RNA structure and protein binding is proposed in the discussion, but the Results section should provide more details about whether efforts have been made to identify the RNA-binding proteins involved. Including information about any planned or preliminary experiments would make the proposed model stronger and more grounded in experimental data.”

Response 3: According the reviewer request, we added the information about potential RNA binding proteins that interact with the NTT region in Discussion.

Comments 4:”In discussion, author suggested that NTT plays a role in regulating splicing in the mod(mdg4) gene. It would be more valuable to add experimental evidence that links NTT to splicing or to discuss relevant literature that supports this function.”

Response 4: In the Discussion section, we have added an overview that provides examples of how canonical termination signals regulate alternative splicing.

Comments 5: “The comparative analysis with D. willistoni offers an interesting evolutionary perspective, but the discussion could further explore how sequence variations between species influence the functional outcome of NTT activity. Are the conserved regions alone sufficient for termination, or do the non-conserved variations play a role in termination efficiency? A more detailed discussion of these sequence differences would add depth to the evolutionary context of the study.”

Response 5: We thank the reviewer for the important comment. In the revision, we have more clearly delineated the role of the conserved sequences in the "Results" section.

Comments 6: “While the proposed mechanism of non-canonical transcription termination is convincing, the Discussion should broaden its scope by connecting these findings to larger themes in transcription regulation. Adding the experimental evidences in the context of known non-canonical termination mechanisms in other species would underscore the importance of the findings in the field of transcription regulation.”

Response 6: In accordance with the reviewer's request, we have expanded the discussion.

Comments 7: “Overall, the manuscript presents an interesting result. Including suggestions for future directions would further enhance its impact and provide valuable insights for similar future research.”

Response 7: We are very grateful to the reviewer for the positive assessment of the work. We have added information about potential partner proteins and research perspectives in the Discussion section.

Round 2

Reviewer 1 Report

Comments and Suggestions for Authors

The authors have properly addressed almost all the concerns raised in the first round of review. My only suggestion would be to add one or two sentence(s) (either in Methods, Results or Discussion), regarding why the authors used a longer construct in in vivo studies although they identified a much smaller region in vitro, to put things in perspective.

Author Response

Comments 1: My only suggestion would be to add one or two sentence(s) (either in Methods, Results or Discussion), regarding why the authors used a longer construct in in vivo studies although they identified a much smaller region in vitro, to put things in perspective.

Response 1: The sentence has been added (marked in blue).